# Water Balance of Pit Lake Development in the Equatorial Region

Edy Jamal Tuheteru [1,2], Rudy Sayoga Gautama [1], Ginting Jalu Kusuma [1,*], Arno Adi Kuntoro [3], Kris Pranoto [4] and Yosef Palinggi [4]

1   Mining Engineering Research Group, Faculty of Mining and Petroleum Engineering, Institut Teknologi Bandung, Bandung 40132, Indonesia; ejtuheteru@student.itb.ac.id (E.J.T.); r_sayoga@mining.itb.ac.id (R.S.G.)
2   Department of Mining Engineering, Faculty of Earth and Energy Technology, Universitas Trisakti, Jakarta 11440, Indonesia
3   Water Resource Engineering Research Group, Faculty of Civil and Environmental Engineering, Institut Teknologi Bandung, Bandung 40132, Indonesia; arnoak@ftsl.itb.ac.id
4   Environmental Department, PT. Kaltim Prima Coal, Sangatta 75611, Indonesia; Kris.Pranoto@kpc.co.id (K.P.); Yosef.Palinggi@kpc.co.id (Y.P.)
*   Correspondence: jaluku@mining.itb.ac.id; Tel.: +62-81223667519

**Abstract:** In recent years, Indonesia has become the largest coal exporter in the world, and most of the coal is being mined by means of open-pit mining. The closure of an open-pit mine will usually leave a pit morphological landform that, in most cases, will be developed into a pit lake. One of the main issues in developing a pit lake is the understanding of the pit lake filling process. This paper discusses the hydrological model in filling the mineout void in a coal mine in Kalimantan which is located close to the equatorial line. The J-void is a mineout coal pit that is 3000 m long and 1000 m wide, with a maximum depth of 145 m. The development of the J-void pit lake after the last load of coal had been mined out experienced a dynamic process, such as backfilling activities with an overburden as well as pumping mine water from the surrounding pits. There are two components in the model, i.e., overland/subsurface and pit area. The overland zone is simulated using the Rainfall-Runoff NRECA Hydrological Model approach to determine the runoff and groundwater components, whereas the pit area is affected by direct rainfall and evaporation. The model is validated with the observation data. The main source of water in the J-void pit lake is rainwater, both from the surrounding catchment area as well as direct rainfall. As this coal mine area is characterized as a multi-pit area and, consequently, several pit lakes will be formed in the future, the result of the hydrological model is very useful in planning the future pit lakes.

**Keywords:** hydrology model; pit lake; equator area

## 1. Introduction

Coal mining activities using the open-pit mining method will leave voids when mining activities end. Mine voids have come to represent a problem for the environment [1] worldwide. The mine void cannot always be backfilled with overburdened material and will leave some openings, which then will be deliberately filled with water to form a mine void lake, which is referred to as a pit lake [2–6]. Pit lakes are distributed around the world and are dominant in Australia, several countries in Europe, and North America [7]. For example, there are more than 1800 mine voids in Western Australia [8], around 100 pit lakes with ages ranging from 24 to 100 years in the Łuk Mużakowa region, Poland [9], and about 30 pit lakes in the Iberian pyrite belt region, Spain [10].

Indonesia is the world's most significant coal producer, with coal production in 2020 at around 500 million tons. Coal production generally comes from mining activities using the surface mining method. Many of those surface coal mines potentially leave voids. The surface coal mine areas, which were chosen as the studied areas in this research, belong to

one of the largest coal mining companies in Indonesia, namely PT. Kaltim Prima Coal. This coal mine operates in two regions, namely Sangatta and Bengalon, and due to the geological characteristics of these areas, it is divided into several pits operated simultaneously. According to the mining closure plan prepared by this coal mining company, with the best option of mining sequence optimization available, several mine voids will inevitably exist at the end of the mine's operation period and will become pit lakes afterward. The company has a liability to make sure that the water quality of these pit lakes meets the government's water quality standards. To fulfill this requirement, the initial characterization step needs to be conducted in order to understand the parameters involved and their influence on the pit lakes' development, which will significantly determine the final water quality of the pit lakes.

Pit lake fill times are site-dependent and disaggregated on a case-by-case basis [11]. Areas with high levels of evaporation will experience a slow filling process. In the pit lake formation process, an essential factor considered is the water balance, which consists of several hydrological components: surface water or runoff water from the surrounding catchment area, direct rainfall into the pit, groundwater, and also the loss of water from the evaporation process [12–14]. These components will influence the hydrological and hydrogeological characteristics of the area and significantly affect the process of forming a pit lake [15,16]. In regions with arid climates, evaporation will be more significant when compared to incoming water; for example, groundwater in the Americas is the most dominant source of water, while in areas with a humid climate, such as in New Zealand, the most prevalent water source is surface water [17].

The hydrological model has been applied to the pit lake formation process previously. Several studies used the Water Budget model [18], such as in the Lusatian district, Germany; the MIKE SHE Model [19], in Pebble Mine Project, Alaska; and the SWAT Model [20], in the Iberian Pyrite Belt mining area, Spain. The TOPMODEL hydrological model was used by researchers in Indonesia to characterize the effect of changes in rain catchment areas on water quality in river areas surrounding mining activities [21]. In Indonesia, there is no research related to the use of hydrological models for the formation of pit lakes. In this study, the calculation of the pit lake filling process was carried out. The calculation process was carried out by dividing the area into two zones, namely the pit lake zone and the overland zone. The pit lake zone used water balance calculations for the pit lake formation process. Meanwhile, for the overland zone, the NRECA hydrology model was used, which can be applied to areas experiencing dynamic changes, while also helping to generate some parameters that are not available in the field, such as groundwater data.

The aim of this research is to obtain a hydrological model for a pit lake formation process in the tropical region. This model can be used to simulate the pit lake filling process, by considering all water balance parameters, starting from the end of mine's pit operation until the water level reaches its equilibrium, which is either stuck at a certain level or can reach the discharged water level. By means of the simulation conducted, it is also expected that the main controlling parameters on forming a pit lake in the tropics can be determined. When the process of the pit lake's formation and its controlling parameters are firmly attained, this model will become one of the important tools for simulating comprehensive pit lake development, both physically and geochemically, to determine the final water quality of the pit lake in the post mine period.

## 2. Materials and Methods

### 2.1. Site Description

#### 2.1.1. Location

The research area is in the mining area of PT. Kaltim Prima Coal (PT. KPC), with geographic boundaries 117°27'7.40''–117°40'43.40'' east longitude and 0°31'20.52''–0°52'4.60'' north latitude, and is included in the administrative area East Kutai District, East Kalimantan Province, Indonesia (Figure 1). The activities carried out include exploration, mining, and marketing activities, with a current coal contract of work (CCoW) area of 84,938 hectares,

including the Sangatta and Bengalon mining areas (the red line in Figure 1), with coal production per year of 70 million tons. The study area of Sangatta lies between the Bengalon and Sangatta river, which flows into the Makassar Strait in the east. The study area's dominant topographical feature is hilly areas, with the highest peak up to +330 m above sea level, which is located in the Dome area. The study area has a tropical climate, with an average temperature of 27.87 °C, relative humidity of 63–100%, an average wind speed of 1.34 m/s, a moderate pressure of 102.77 kPa, and an average length of irradiation of sun of 5.28 h/day.

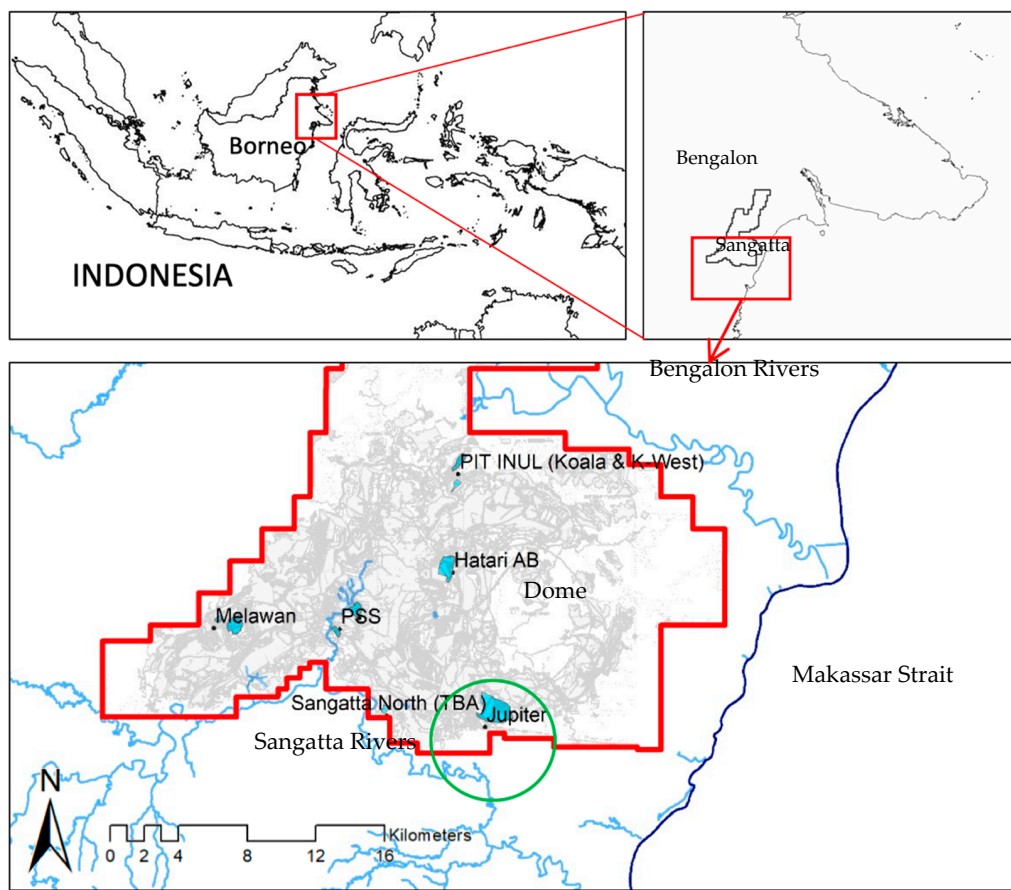

**Figure 1.** Study area.

### 2.1.2. Geology and Morphology

The Sangatta area is located between the Mahakam Delta and Tinggian Mangkalihat, which are part of the northern Kutai Basin. The research area itself is composed of the Balikpapan Formation, with a lithology of sandstone, claystone, silt, tuff, and coal. In the alternation of quartz sandstone, clay and silt, a cross structure is formed [22]. Locally, coal inserts are present with a thickness of between 20 and 40 cm. The clay in the area is gray, brittle, and contains muscovite, bitumen, and iron oxide (Figure 2). The thickness of the formation is ±2000 m, with a land-delta frontal depositional environment. This formation can be dated to the Middle–Late Miocene period [23]. In general, the morphology of the Kutai Basin can be divided into three units, namely steep hills, sloping hills, and plains. Steep hills are generally found on the edges of the basin with steep reliefs. Sloping hills with wavy reliefs are generally found in the middle of the basin. The morphology of the plains is indicated by the area without hilly reliefs located in the middle of the Muarakaman area up to the Mahakam delta (Samarinda). The research area is located in an area with a sloping hilly morphology.

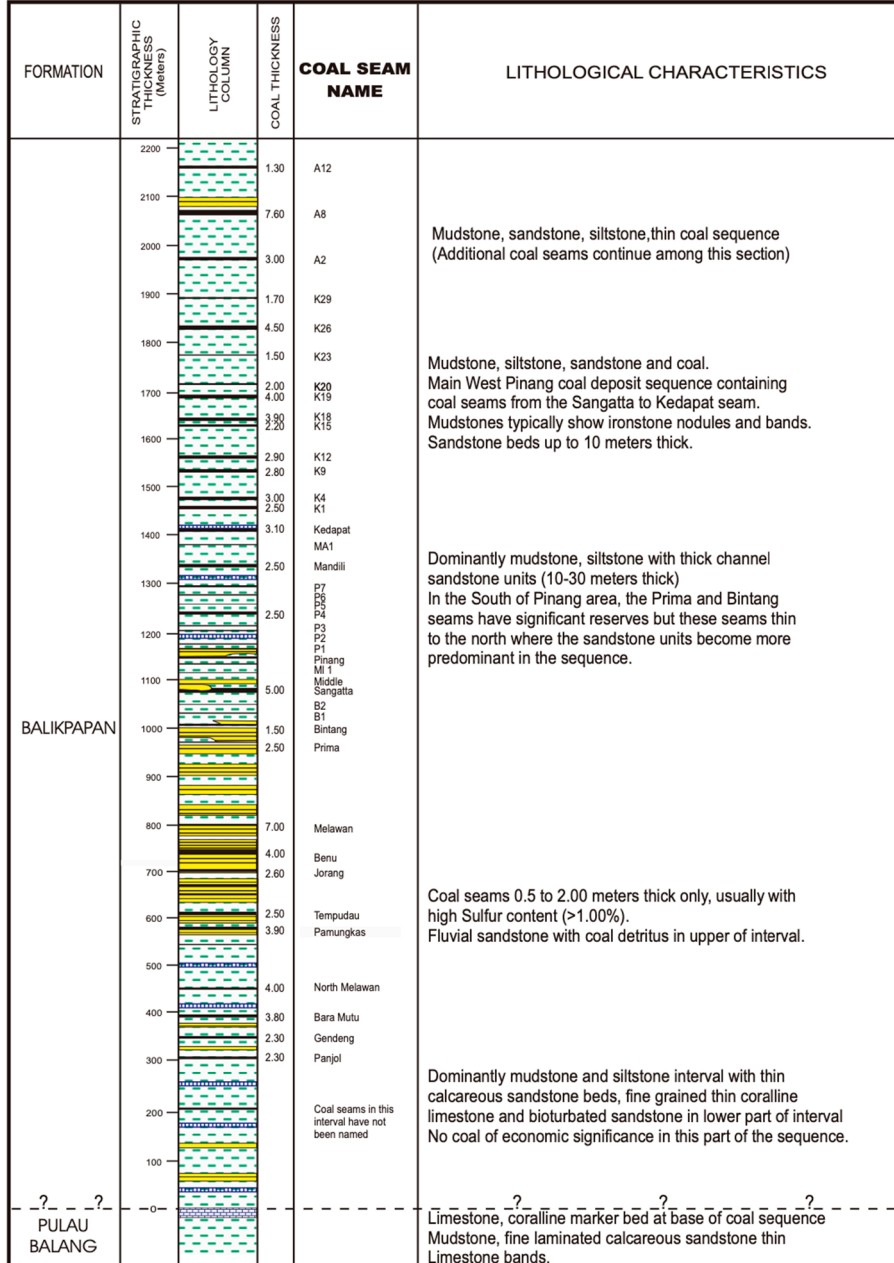

**Figure 2.** Stratigraphy of study area.

### 2.2. Calculation of Water Balance

Water balance calculations were used to determine the amount of water flowing into and discharged out of the pit lake in the formation process. By understanding these components' or parameters' relationships and behaviors, the prediction of how long it will take time to reach an equilibrium condition can be conducted. Parameters that constituted water inflow were direct rainfall, runoff water, and groundwater entering the pit lake. In contrast, water discharge parameters included evaporation and groundwater coming out of the pit lake. In addition to the main parameters, water inflow also came from the surrounding catchment area of the pits, from surface water surrounding the river, or from pumping water from nearby pits in some cases. The water balance equation [24]:

$$\delta V_L = Vp + V_R + V_{GW} + V_{pm} - V_E \tag{1}$$

where $\delta V_L$ represents the changes in pit lake volume per period time (m$^3$), $Vp$ is the volume of direct precipitation (m$^3$), $V_R$ is the volume of runoff water (m$^3$), $V_{GW}$ is the volume of groundwater (m$^3$), $V_{pm}$ is the water pumping volume, and V$_E$ is the volume of water lost, i.e., volume evaporation (m$^3$).

Calculation of the water balance in the pit lake consisted of two zones, namely the overland/subsurface zone and the pit lake zone. Figure 3 shows that the overland/subsurface zone calculation using the hydrological model of NRECA with the involving factors, namely direct rainfall, and the potential for evapotranspiration, which will determine the runoff and groundwater parameters. The influencing factors of the pit lake zone included the direct rainfall and evaporation potential, which govern the water balance in the pit lake water body.

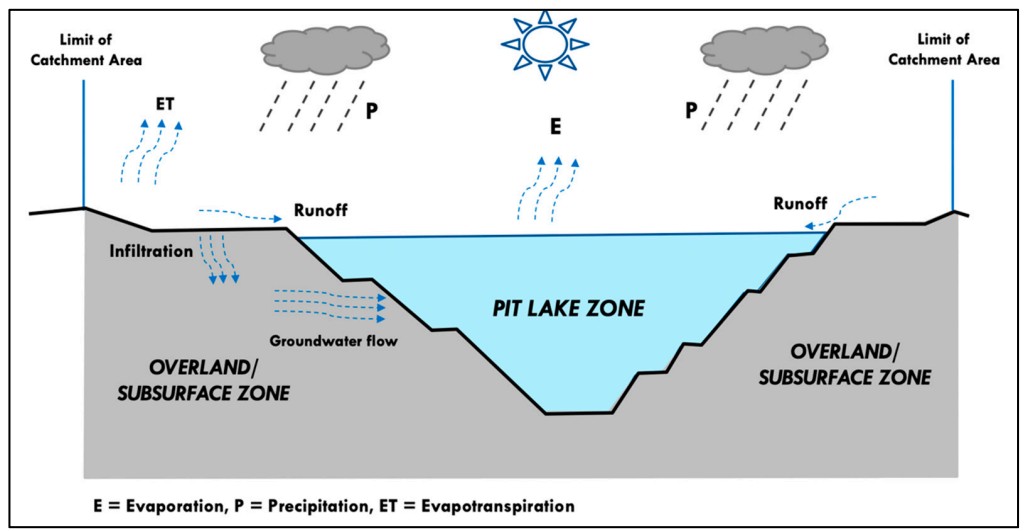

**Figure 3.** Conceptual water balance calculation.

2.2.1. Overland/Subsurface Zone

The rainfall-runoff NRECA (National Rural Electric Cooperative Association) Model was used to simulate the overland/subsurface zone. Crawford and Thurin developed this model in 1981 based on the water balance equation [25]. The rainfall flows over the surface, and the base flow moves into the river channel. The total flow that exists is then multiplied by the area of the watershed. The result of this multiplication is the output of the NRECA model in the form of river flow discharge according to the planned period. This model can be used to calculate the monthly discharge of monthly rainfall based on the water balance in the watershed [26]. The NRECA model divides the monthly flow into direct runoff (surface and subsurface runoff) and baseflow (Figure 4).

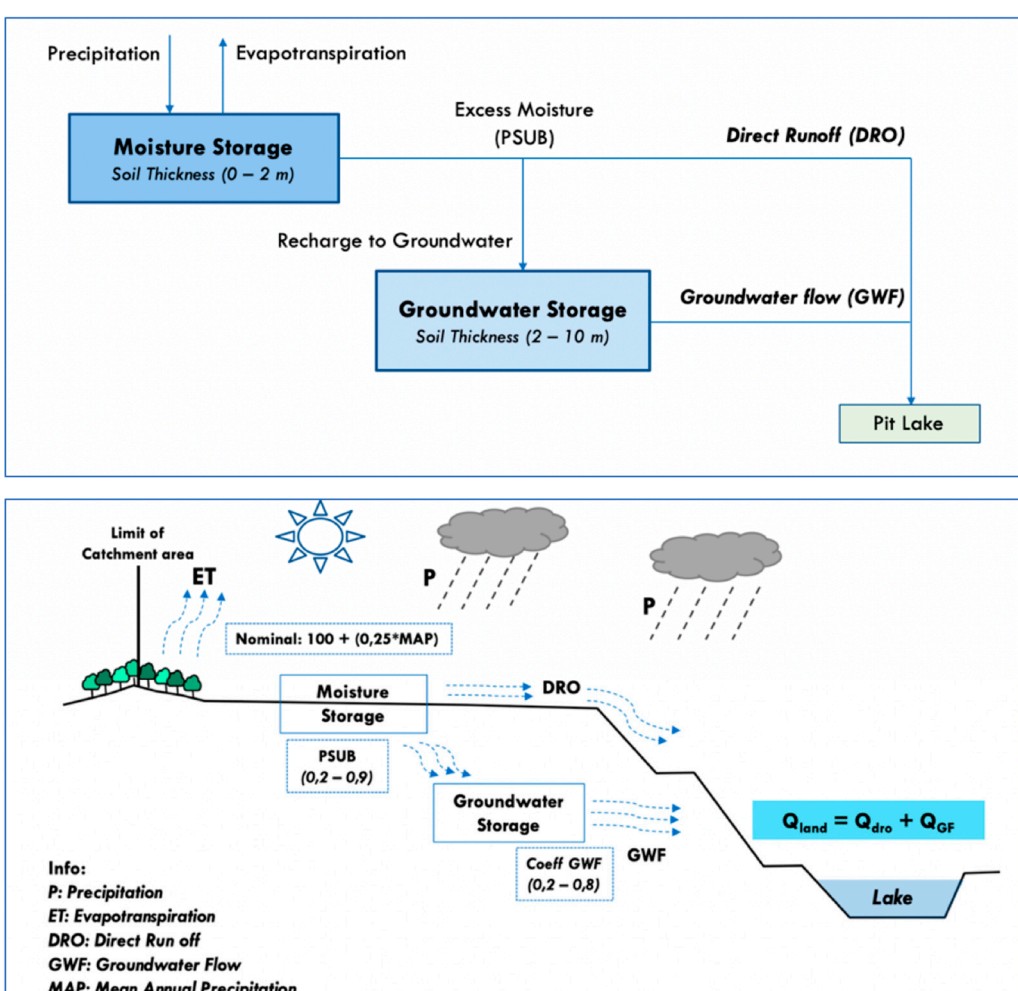

**Figure 4.** Overland zone—NRECA model.

Three or four parameters were used in this model to generate discharge data on the basis of the monthly rainfall data. The NRECA model structure then divided the monthly flow into runoff and groundwater storage parameter data (Figure 4). The NRECA parameter calibration stage was conducted in order to find the most suitable parameters and the closest to the actual conditions. Parameters of Model NRECA:

1.  Moisture Storage = Initial moisture storage value This value was entered by trial and error and was re-checked so that the value in January approaches the value in December. If there was a difference greater than 200 mm, it was repeated. The value of Moisture storage is 1000.
2.  Nominal = Index of moisture holding capacity Nominal = 100 + (C ∗ Average Annual Rainfall), C value = 0.2 for areas with year-round rainfall, C value = 0.25 for areas with seasonal rainfall. The average annual rainfall was 2062 mm, and so the value of Nominal was 616.
3.  PSUB = percentage of surface runoff that enters the groundwater reservoir PSUB = 0.5, for watersheds with normal/ordinary rain. PSUB = 0.5 < PSUB < 0.9, for areas with large permeable aquifers. PSUB < 0.5, for areas with limited aquifers and thin soil layers.
4.  Begin Store GW = Initial storage of groundwater Performed manually and starting with an initial value of 300.
5.  Ground Water Flow (GWF) = groundwater flow rate. GWF = 0.5; for areas with normal/normal rain. GWF = 0.5 < GWF < 0.8; for areas that have a continuous flow with a small size. GWF = 0.2 < GWF < 0.5; for areas that have reliable continuous flow.

6. Kc = crop coefficient, with a value 0.36.

This modeling is based on a spreadsheet program [25]. The water balance calculation results estimate the percentage of the potential availability of surface water and groundwater around the pit lake. The results of these calculations are used to create a hydrogeological model that describes the water distribution pattern. Furthermore, these results are also used to calculate the volume of water in the pit lake filling stage.

### 2.2.2. Pit Lake Zone

The water volume calculation in the pit lake zone was influenced by direct rainfall and evaporation, where an increase in volume came from rainfall while a decrease in water volume was due to the evaporation process. The water volume change in the pit lake zone was obtained by performing calculations, as shown in Figure 5. The addition of water volume from rainfall was the product of the multiplication of the monthly rainfall by the pit lake's surface area. This calculation was carried out from the beginning of the water filling process of the pit lake in the first month right after the mining operation had stopped until the discharge level was reached. Similarly, the calculation of water loss was conducted by multiplying the value of evaporation by the area of the body of water in the pit lake at every stage/phase until the discharge level was reached. Measurement of the volume of water in the pit lake was carried out by the survey team by measuring the position of the water level at a certain elevation, and then, with the help of Minex 6.5 software, they determined the volume of water, which is the difference between elevations [20].

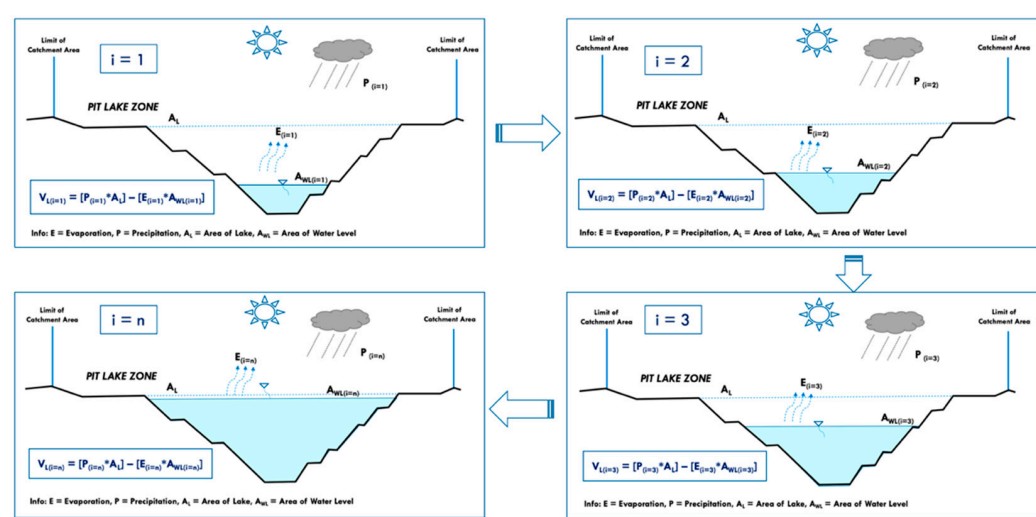

**Figure 5.** Pit Lake Zone.

### 2.3. Data Collection

#### 2.3.1. Rainfall Measurement

There are 12 rainfall gauge stations scattered throughout the CCoW area for the Sangatta Block, most of which are manual gauges. However, in recent years, several automatic rainfall gauges were installed. Data from manual gauges are collected simultaneously around 08:00 to 10:00 a.m. central Indonesian time (+8 GMT to +10 GMT). From the twelve available rain gauges, one closest gauge was used for rainfall-runoff analysis. The rainfall data series has been available since 2005.

#### 2.3.2. Potential Evapotranspiration and Evaporation Calculation

Potential evapotranspiration (PET) was calculated using the Penman–Monteith method. The Penman–Monteith method was also applied to the calculation of potential evapotranspiration in several pit lake areas, such as in the Czech Republic [27,28]. This evapotranspiration value will be used as an input to the Overland Zone Model. The input data for PET calculation are the air temperature, humidity, duration of sun exposure, wind speed,

elevation, and geographical latitude of the study area. Climatology data for PET calculation were obtained from the Meteorology, Climatology, and Geophysical Agency (BMKG) of Samarinda, located about 123 km from the study area. The formula for calculating PET using the Penman–Monteith method [29] is as follows:

$$ET_o = \frac{0.408 \Delta R_n + \gamma \frac{900}{(T+273)} U_2 (e_s - e_a)}{\Delta + \gamma (1 + 0.34 U_2)} \tag{2}$$

where $ET_o$ is the potential evapotranspiration (mm/day), $\Delta$ is the slope of the water vapor pressure curve to air temperature (kPa/°C), $\gamma$ is the psychrometric constant (kPa/°C), $T$ is the mean air temperature (°C), $U_2$ is the wind speed at an altitude of 2 m above the ground (m/s), $e_s$ is the saturated water vapor pressure (kPa), $e_a$ is the actual water vapor pressure (kPa) and $R_n$ is the net aboveground solar radiation (MJ/m$^2$/day).

The evaporation estimate was obtained using Meyer's Formula [30]. This evaporation value is then used as an input to the Pit Lake Zone Model.

$$E_L = K_M (e_w - e_a) \left[ 1 + \frac{U_9}{16} \right] \tag{3}$$

where $E_L$ = lake evaporation (mm/day), $e_w$ = saturated vapor pressure at the water surface temperature (mm of mercury), $e_a$ = actual vapor pressure of over-lying air at a specified height (mm of mercury), $U_9$ = monthly mean wind velocity (km/h) at about 9 m above ground and $K_M$ = coefficient accounting for various factors with a value of 0.36 for large amounts of deep water and 0.50 for small amounts of shallow water.

### 2.3.3. Overburden Backfilling and Water Inflow

There were two mining activities that took part in influencing the pit lake's development, namely backfilling and water pumping. Backfilling is an activity that involves placing the overburdened material back into the mine void rather than placing it at a dump outside of the pit, as required by the Indonesian government to minimize the mine void as a part of the good mining practices campaign. This activity will reduce the capacity of the final pit lake compared to the initial mine void after the mining operation is stopped. The water pumping, in turn, includes water pumping from another pit sump and pumping from the sediment ponds surrounding the J-void. The water pumping from the pit sump was continuously conducted with huge capacity, while the pumping from the sediment ponds was conducted occasionally, once or two times a year when the sediment ponds needed to be maintained.

## 3. Results

### 3.1. Geometry of Void

Pit lake J-void was a former mining site called Jupiter pit belonging to PT. Kaltim Prima Coal. Mining operations at Jupiter pit started in 2004 and ended after ten years of mining operations in 2014. At the end of mining operations, the geometry of the J-void was 1500 m wide, 3500 m long, and a 145 m maximum depth. The total area at an elevation of +13 m asl, which became the final water body, was 2.89 km$^2$, with a total water capacity of 180 million m$^3$. The catchment area of the pit lake was 21.43 km$^2$, which mostly consisted of the reclamation area (Figure 6).

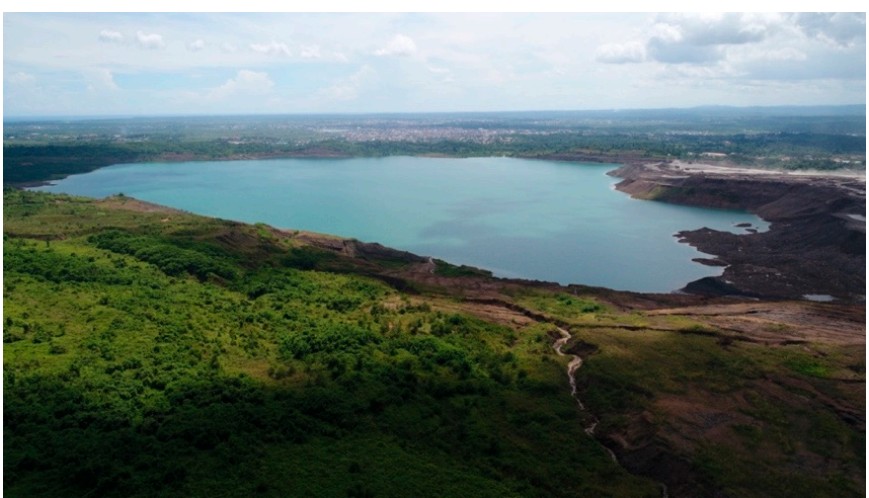

**Figure 6.** Actual Pit Lake J-Void.

*3.2. Backfilling*

Backfilling activities were carried out on the western part of the J-void. Figure 7 shows changes in the J-void shape, especially on the west side, which had the lowest elevation in the J-void, at −145 m bsl. The backfilling material came from the Bendili pit, which lies near the J-void. The backfilling influenced both the capacity of the pit lake and the surface/catchment ratio between the overland/surface zone and the pit lake zone. The yearly changes in the ratio year to year in 2014, 2015, 2016, and 2017 were 8.5, 8.8, 9.93, and 10.18, respectively. Along with the backfilling process, water filling began in September 2014, until it reached the discharge level at the end of 2017.

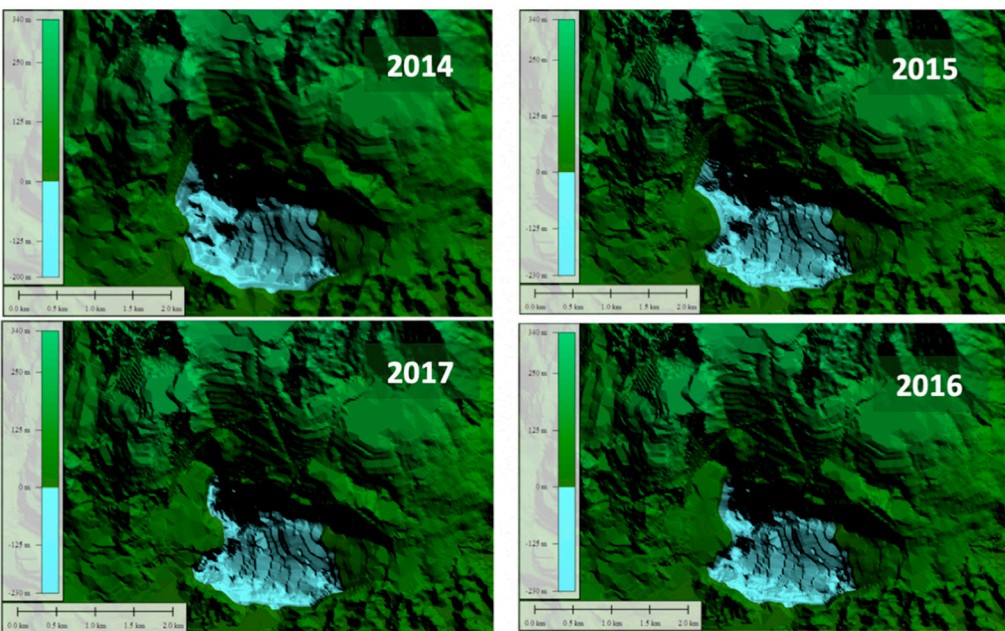

**Figure 7.** Backflling.

*3.3. Rainfall*

The rainfall data measured from 2014 to 2017 in the study area are shown in Figure 8. The average annual rainfall was 2162.4 mm, the highest annual rainfall was 2521.5 mm (occurred in 2017), and the lowest annual rainfall was 1593.5 mm (occurred in 2015). The wet period occurred from November to May, and the dry period was from June to October.

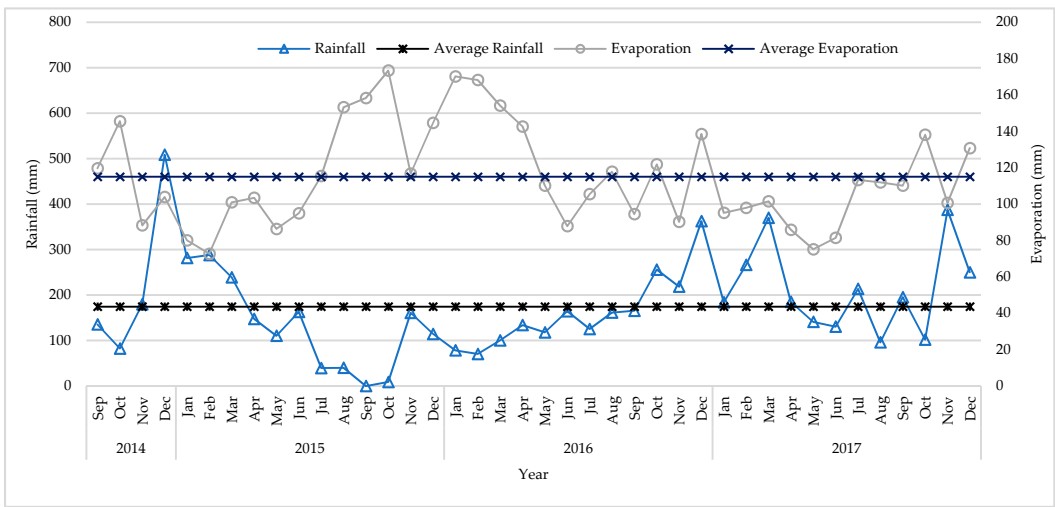

**Figure 8.** Total monthly rainfall and evaporation.

In terms of monthly rainfall-based data, the lowest monthly average was 132.79 mm, which occurred in 2015, which had the lowest maximum rainfall as well as minimum rainfall, with values of 288.0 mm/month and 0.0 mm/month, respectively. In addition, the maximum monthly rainfall did not occur in the year with the highest annual rainfall; rather, it occurred in December 2014, with a rainfall of 508.5 mm/month, which shows the variability of monthly rainfall in the study area.

### 3.4. Evaporation

The evaporation value was calculated using Equation (3) based on the climate data obtained, including air temperature, humidity, sun radiation duration, wind speed, the latitude of the study area, and elevation of the study area. The data were collected from the East Kalimantan in Numbers Report of Years 2014 [31], 2015 [32], 2016 [33], and 2017 [34]. In 2014, the lowest value was 88.30 mm/month, the highest was 145.54 mm/month, and the average was 114.36 mm/month. In 2015, the lowest value was 72.72 mm/month, the highest was 173.42 mm/month, and the average was 116.71 mm/month. In 2016, the lowest values were 87.89 mm/month, the highest was 170.19 mm/month, and the average was 125.14 mm/month, while in 2017, the lowest value was 75.17 mm/month, the highest was 138.13 mm/month, and the average was 103.52 mm/month. Overall, the highest evaporation value was obtained in October 2015 at 115.05 mm, and the lowest occurred in February 2015, which was 72.72 mm (Figure 8).

### 3.5. Pumping Water

There was a mining pit that was still in the operating stage, namely, the Bendili pit, and two sediment ponds, namely, the Kenny J and Azalea ponds, around the study area. The pit sump water in the Bendili pit was pumped directly into the J-void pit lake, and occasionally slurry from Kenny J and Azalea ponds was pumped into the J-void pit lake. The first pumping was carried out in August 2015, a year after the mining activity ended. In 2015, the highest pump discharge occurred in November, at 941 thousand m$^3$, and the lowest occurred in August, at 178 thousand m$^3$; in 2016, the highest pump discharge occurred in November, at 2.5 million m$^3$, and the lowest occurred in June, at 119 thousand m$^3$; and in 2017, the highest pump discharge occurred in December, at 3 million m$^3$, and the lowest occurred in August, at 609 thousand m$^3$. The total pumped water from August 2015 to the end of 2015 was 2,055,000 m$^3$; in 2016, it was 12,208,000 m$^3$; and in 2017, it was 23,680,000 m$^3$ (Figure 9). During the filling period, the pump discharge ranged from 119.2 thousand to 3.0 million m$^3$, with a total of 38.4 million m$^3$ over that period. Water pumping shows that the study area increased every year until finally experiencing an overflow in late December 2017.

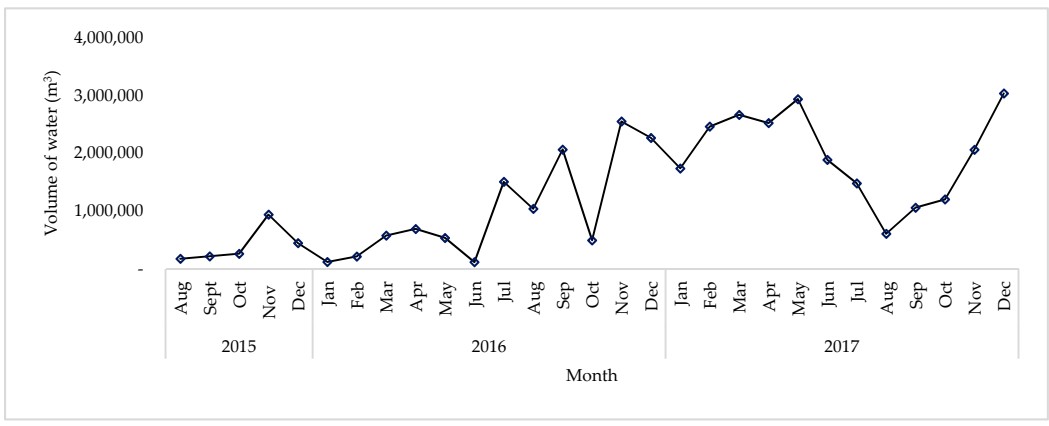

**Figure 9.** Pumping water from Bendili pit.

*3.6. Water Level*

The water level in the pit lake is measured by mapping the water level The measurement results show that the water level in the pit lake at the beginning of its formation is 120 mbsl. At the beginning of 2015, the water level was 82 mbsl; at the end of 2015 or early 2016, it was 56 mbsl. At the beginning of 2017, the water level was 11 masl. The water level reached the output at the end of 2017, at an elevation of 13.5 masl (Figure 10).

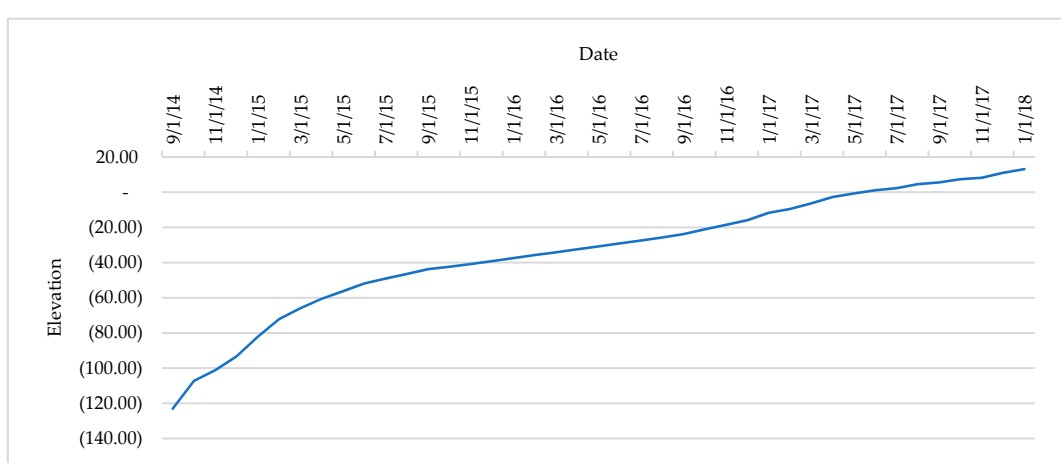

**Figure 10.** Water level.

*3.7. Water Balance*

As mentioned above, the water flowing into the pit lake during the pit lake formation stage came from the overland/subsurface zone, the pit lake zone, and direct pumping from another pit or sediment ponds. In the overland/subsurface zone, calculation was performed using the NRECA model. The catchment area in the overland zone changed, along with overburden material filling in the voids: in 2014 it was 21.46 km$^2$, in 2015 it was 21.58 km$^2$, in 2016 it was 21.87 km$^2$, and in 2017 it was 21.93 km$^2$. The nominal parameter was multiplied by the coefficient of 0.25 since the study area was a categorized area with seasonal rainfall. For the PSUB parameter, a value of 0.35 was used, considering the stable soil conditions and a minor water release. A value of 0.22 was used for the GWF, considering the groundwater level at a depth of 2–10 m below the surface. For the crop coefficient, a value of 0.36 was used, considering the reclamation area condition with vegetation. The water inflow coming from the pit zone was calculated by multiplying the monthly rainfall by the pit lake's surface area, whereas the direct pumping volume from

the Bendili pit was collected according to company measurements. The calculation results of these water inflow parameters from late 2014 to late 2017 are shown in Figure 11.

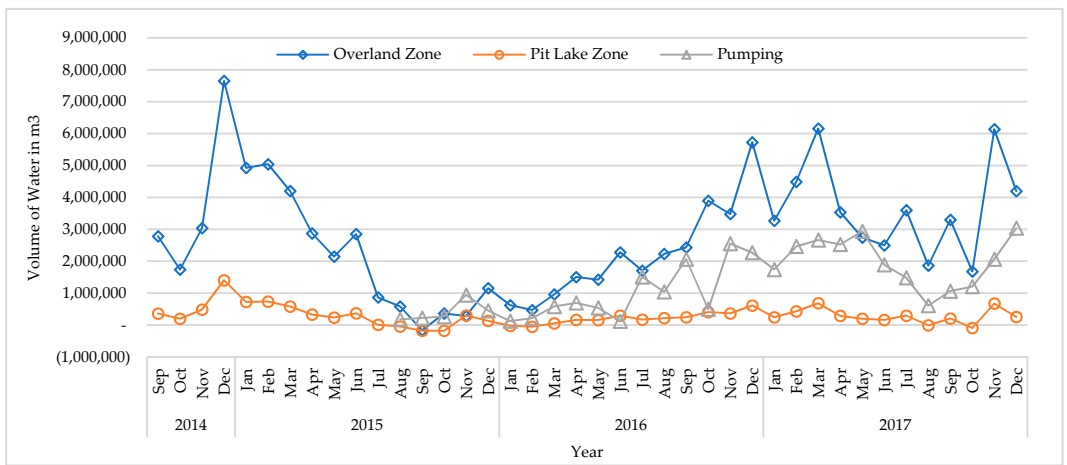

**Figure 11.** Water Balance.

The most significant water volume in pit lake formation came from the overland/subsurface zone, followed by pumping, and finally from the pit lake zone. The water volume from the overland/subsurface zone fluctuated during pit lake development, ranging from 289.1 thousand to 7962.8 thousand m$^3$. There was a period in 2015 where the water inflow from the overland/subsurface zone was low as a result of low rainfall, which was considered to be influenced by the global El Nino effect [35]. The monthly water pumped from the Bendili pit also fluctuated, ranging from 119.2 thousand to 3.0 million m$^3$, with an increasing trend. The increasing trend, despite being related to the increase in rainfall, was also due to operational reasons in the Bendili pit, as the pit area was increased. Water from the pit lake zone was the lowest compared to other parameters mentioned previously since the water volume coming from the pit lake zone was controlled by direct rainfall and evaporation from the pit lake zone, where the yearly water budget surplus was only 629 mm/year or 44%/year.

Figure 12 shows that at the beginning of the formation in 2015, the dominant water volume was contributed by runoff, at around 44%, while that from groundwater constituted 39% of the total, direct rain contributed 6%, pumps contributed 7%, and water loss due to evaporation represented 4% of the total water volume. In 2016, runoff contributed the dominant volume of water, at 47%, while groundwater decreased to 17%, direct precipitation immediately decreased to 4%, pumps increased to 29%, and water loss due to evaporation decreased to 2%. Meanwhile, at the end of the year, with the formation of the pit lake, the runoff's contribution to the water volume was still dominant at 40%, while groundwater contributed around 21%, direct precipitation reduced to 3%, pumps continued to increase to 34%, and evaporation remained at 2%.

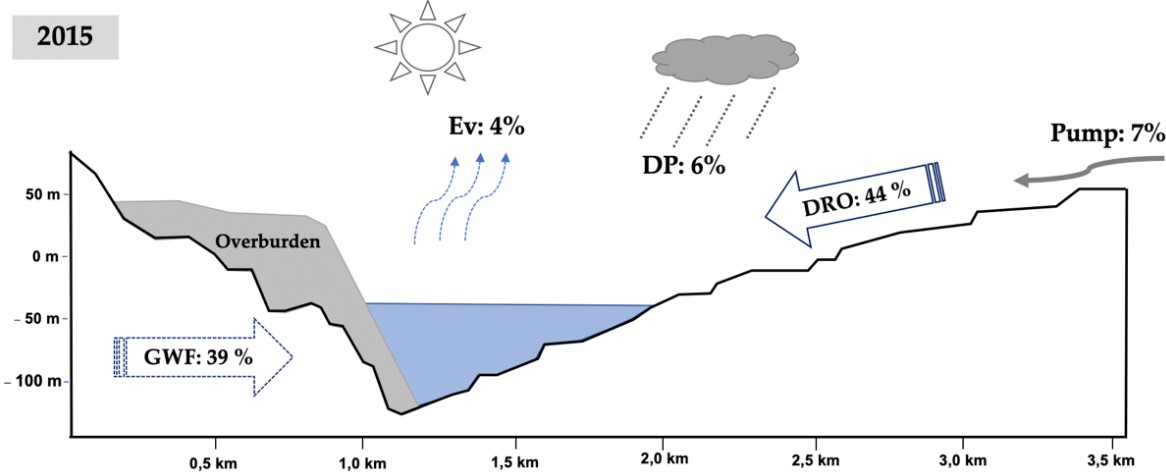

Ev: Evaporation, DP: Direct Precipitation, DRO: Direct Runoff, GWF: Groundwater flow

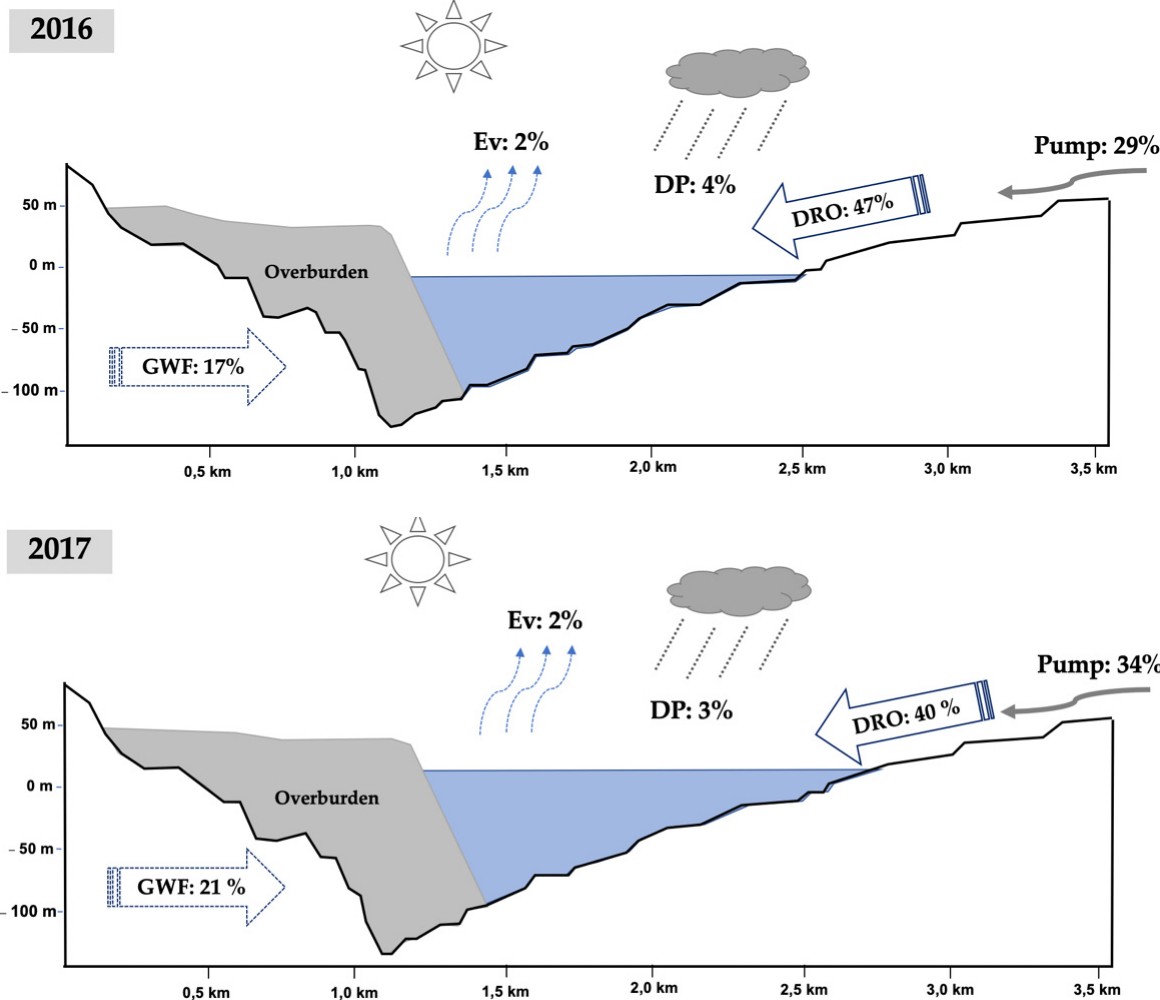

**Figure 12.** Results of the pit lake formation simulation model.

The runoff's dominant contribution to the volume of water was due to the influence of the extent of the attachment area, which continued to increase every year. The initial ratio of 1:8 became 1:10 in 2017. As the catchment area increased, it also affected the runoff water that entered the pit lake. Another factor that was also very significant in forming this

pit lake was the activity of water pumping, which showed an increase every year, from 7% at the beginning of the year to 34% in the final year.

Direct precipitation entering the voids did not have a massive effect; this is because the left mine voids' dimensions decreased due to the filling of mine voids with rocks, thereby reducing the surface area of the pit lake. In 2015, water replenishment was around 6%, and in 2017, it was about 3%. Loss of water volume occurred during the evaporation process; according to Figure 13, evaporation was more significant in 2015, during the dry season. From July to October, the rainfall was deficient. When climatic conditions were wet or precipitation was high, water loss diminished.

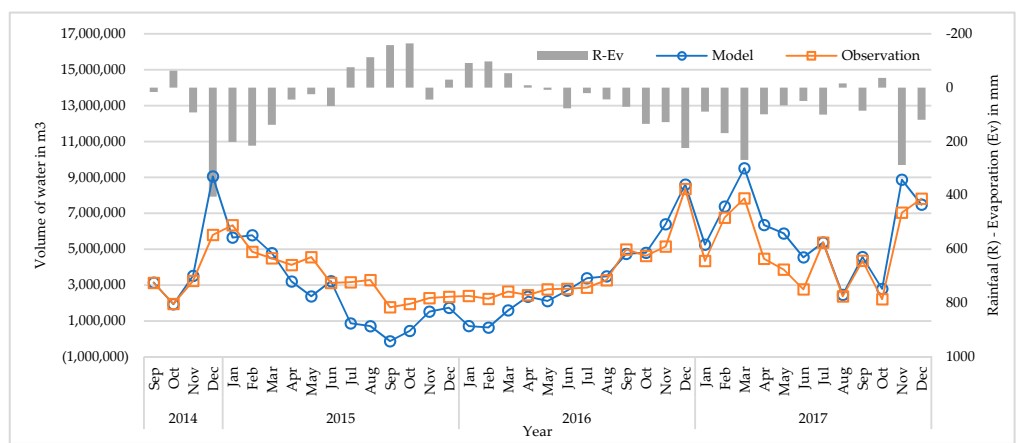

**Figure 13.** Simulation model validation.

Based on the water filling model's simulation results in the pit lake formation process, the most influential component of the pit lake formation is runoff water. The cause of the ample runoff water is the vast catchment area; the more significant the catchment area, the faster the pit lake's filling process will be. The surface water's dominance is the same as in New Zealand and Papua New Guinea, where the surface water volume is greater than that of the groundwater [17].

### 3.8. Comparison, Validation, and Sensitivitas Analysis of Model

Figure 13 shows the results of the actual volume of water when filling the pit lake. The cumulative volume of water in the filling process continued to increase from September 2014 to December 2017, with the total volume of water reaching an equilibrium of 104 million m$^3$. The obtained results, including the comparison between the actual measurements and the simulation model that was developed, are shown in Figure 13 It can see that the model simulation results have the same trend as the observation results from the beginning of the filling process up to June 2015. After that, there was a difference of about 8% in the water's cumulative volume. This difference is due to filling the pit lake with water from another pit with no input in the simulation model. The process of filling observations is faster than filling using a simulation model. Furthermore, June 2017 shows the same relationship between the simulation model and the observations.

We validated the model by calculating the NSE value [36] and the correlation coefficient [37]. The NSE value shows how well the plot of the observation versus the model simulation fits the 1:1 line. The NSE ranges between −∞ and 1.0 (1 inclusive), with NSE = 1 being the optimal value. Values between 0.0 and 1.0 are generally viewed as acceptable levels of performance. An efficiency less than zero (NSE < 0.0) indicates when the observed mean is a better predictor than the model, which indicates an unacceptable performance [38–40]. An NSE nearer to 1 suggests a model with more predictive skill. The correlation coefficient is calculated using Pearson's equation [41], which is the correlation between two variables, in this case, the observation variable and the estimated model. Based on the calculation results, the NSE value was 0.50, with a coefficient correlation of

0.90. According to the NSE value, it can be stated that this model is acceptable, while the correlation value results show a strong relationship [42,43].

Sensitivity analysis used linear regression to find the most dominant factor in the pit lake filling process, using monthly discharge data to obtain a linear regression [44] image, as shown in Figure 14. In Figure 14a, it can be seen that for additional parameters, the volume of water from direct rainfall in the pit lake obtained a total discharge of around 8 m$^3$/s. In Figure 14b, it can be seen that the additional volume of water comes from runoff water, which has a total discharge in the range of 40 m$^3$/s, while Figure 14c shows a reduction in the volume of water that comes from the potential evapotranspiration value, with a total discharge amount of about 5 m$^3$/s. Based on the description, it can be said that the volume of water in the pit lake is dominated by water sourced from runoff, while the water loss due to the evaporation process is not very significant. Direct rainfall and evaporation are sensitive to climate and increase with time, while runoff water is sensitive to the seasonal changes that occur both during the wet and dry seasons. During the wet season, the volume of water increases, and in the dry season, there is a decrease in the volume of water.

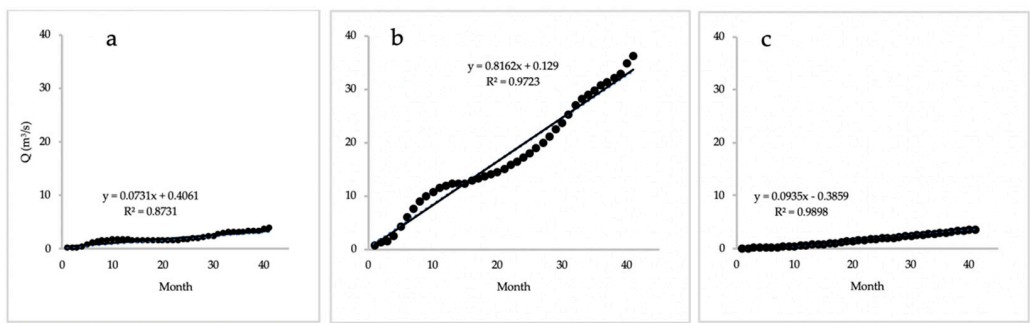

**Figure 14.** Linear regression analysis. (**a**) Direct rainfall, (**b**) runoff and (**c**) evaporation.

## 4. Discussion

### 4.1. Hydrological Characteristics

With regard to pit lake development, hydrological characteristics occupy a dominant role, since they control the rate of water filling and maintain the water level in the pit lake throughout the year. Figure 15 shows the hydrological balance by means of the annual rainfall and annual evaporation ratio of the study area as compared to several areas around the world where most pit lakes exist. The hydrological balance in the study area indicated a surplus of 44%, whereas, in Europe [45], Australia [8,46], and North America [47], the surpluses were 76%, −81%, and −66%, respectively. The hydrological balance in the study area was lower than in Europe but much higher than the hydrological balance in Australia and North America. Although the hydrological balance in the study area was lower than in Europe, the amount of water surplus was much higher, at up to 629 mm/year, than in Europe, which has a value of 450 mm/year. This surplus condition means that the hydrological characteristics of the study area are unique and different from those of other regions and hence represent an important contribution to pit lake development and management.

According to the data for monthly rainfall and monthly potential evapotranspiration from 2014 to the end of 2017, rainfall was quite dispersed throughout the year, relating to the wet period from November to March and the dry season in the period from May to October. The highest rainfall in the wet season, up to 491 mm/month, occurred in December 2014, and the lowest rainfall, approaching 0 mm, was observed in September 2015. The calculated evaporation showed a value range of 72.72–173.42 mm/month. The highest evaporation value occurred in October 2015, in line with the decrease in rainfall for the same period, while the lowest evaporation value occurred in February 2015. The average annual rainfall and annual evaporation values from 2014 to 2017 were 2064 mm and 1237 mm, respectively. Looking at the data for rainfall and evaporation, it is clear that there will be additions and subtractions in the process of pit lake formation.

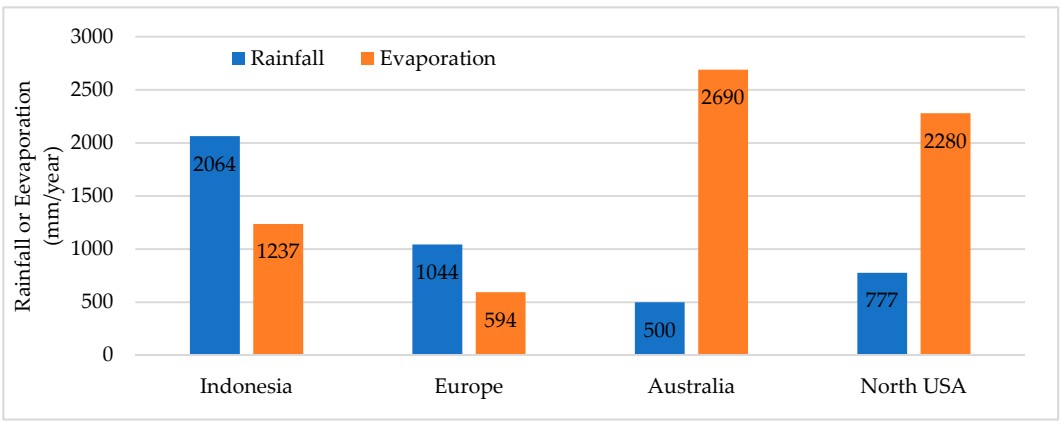

**Figure 15.** Average annual rainfall and evaporation.

*4.2. Pit Lake Development*

According to the elevation, the company measured the development of the pit lake's actual water level using the mapping survey method. After obtaining a specific elevation, we conducted computational calculations using the formula described in [20] to obtain the total water volume in the pit lake. The formation of the main pit lake comes from surface water and pumping from sumps and ponds in the surrounding pits. All pit lake formation components were characterized to obtain information on the main components that affect the process of pit lake formation. Along with the development of pit lake filling, backfilling activities also occurred in the surrounding pits. The addition of rock volume in the pit affects the pit's dimensions [48], which continues to increase the catchment area. In addition to increasing the catchment area, it also affects the volume of surface water entering the pit, where the groundwater volume is slightly reduced. Pit filling uses water and rocks from surrounding pits to speed up the filling of the pit lake and this affects the quality of the water that is being formed [49].

This survey of the water level continued continuously until the pit lake reached equilibrium. Figure 16 shows the changes in the filling of the pit lake from its initial formation in September 2014 until the end of 2017. The water volume addition in the pit lake is obtained by measuring the area each month. The change in the area results in an increased water volume each month from the beginning of the formation to the water surface and outflow elevation. All changes that occur, whether they result in an increased volume of water, a reduced volume of water, or an increase in the volume of rock, are depicted on the mass change curve, which continuously updates with each development.

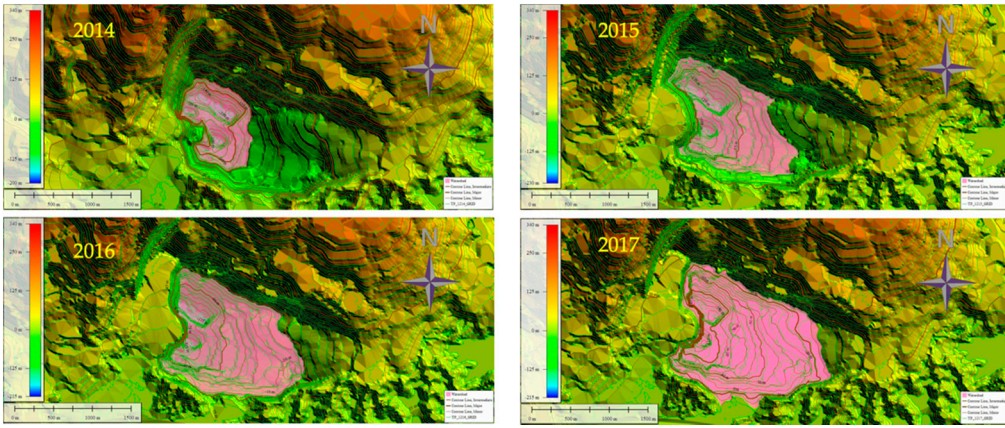

**Figure 16.** The development of a pit lake process.

Inflow into the pit lake comes from runoff and pumps. The total volume of water in the form of natural flows (surface runoff and groundwater) is calculated from changes in the volume of the pit lake minus the total monthly pump volume. There is a typical difference in discharge patterns between those dominated by surface runoff and groundwater. The flow dominated by surface runoff tends to have a high peak flow (discharge during the rainy season) and a low flow (discharge during the dry season). The flow dominated by groundwater tends to have a lower peak flow but a higher low flow. This monthly discharge fluctuation was used to calibrate the PSUB coefficient (the proportion of adequate rain that flows into the surface runoff) and the GWG coefficient (the proportion of stored water in the soil that ends up on the surface as groundwater) in the NRECA model. Based on the calibration, the proportion of water originating from surface runoff and groundwater can be estimated.

### 4.3. Implication of Model

Indonesia is a tropical region with characteristically high rainfall. In the process of filling the pit lake in the study area, it is clear that both runoff and direct rainfall contribute significantly to the increase in air volume. There are also conditions that differ from other areas, suggesting that pit lakes in the tropical area may have a shorter filling time compared to non-tropical areas. The sensitivity results obtained reinforce that water runoff is very sensitive to the process of pit lake formation. The high volume of water originating from runoff is due to the extensive catchment area.

The water loss in the primary pit lake formation process comes from the evaporation component. This study uses data on evaporation due to the absence of direct evaporation measurements. The calculation of evaporation uses actual data released by the Meteorological, Climatological, and Geophysical Agency for the Samarinda region. A loss of air volume from the evaporation component occurs in the dry season from June to October. In total, the loss of the evaporation water volume is not too significant, as seen from the small value produced for the loss of water from the evaporation component. One of the characteristics of tropical areas is that they have low evaporation values.

Other components that contribute to forming a pit lake are pumping and the existence of backfilling activities. Multipit mining, such as in the study area, allows water pumping and backfilling from pits around the pit lake. This pumping activity reduces the amount of water in the active pit and accelerates the filling of the pit lake. Moreover, if the pumped water is of better quality, the low-water-quality formation can occur. Backfill is carried out into the pit lake due to insufficient land availability for the overburdened material. This backfilling plays a role in reducing the dimensions of the pit lake to speed up the process of filling the pit lake. In addition to reducing the dimensions of the pit lake, it also affects the process of groundwater and water absorption in the pit lake.

Another component that is no less important in forming a pit lake is the size of the catchment area. The catchment area is part of the pit lake [50]. The wider the catchment area, the more air runoff occurs and the larger the volume of the air runoff. If the catchment area is small, the water runoff volume will be low. In the study area, which has a large enough catchment area, the volume of water runoff became a component of the formation of such a large pit lake. In the end, it also sped up the pit lake-filling process.

As already mentioned, the components of lake formation include climate local conditions, hydrology, hydrogeology, pumping, backfilling, and the size of the catchment area. All of these components influence the process of pit lake formation. The characterization of all components of pit lake formation is essential for the development of hydrological models. The hydrological model developed in this research area is confirmed to be close to the measurement model and well-validated. By utilizing the hydrological model created in this study, it is possible to predict the time of pit lake formation. By knowing the time of pit lake formation, pit lake management activities can be applied.

## 5. Conclusions

The main components that affect the pit lake formation process are the shape or dimensions of the voids, the catchment area, and the hydrogeological conditions. This study shows that in terms of the water balance during the formation in 2014 to reach equilibrium in 2017 with the addition of the volume of water comes from; 40–47% direct runoff, 17–39% groundwater flow, 3–6% direct precipitation, and 7–34% pumping, while the reduction in the volume of water comes from evaporation which ranges between 2–4%. The process of filling a pit lake with water, which continues until a state of equilibrium is reached. Based on the previously described analysis, the most dominant factor in the pit lake's filling process is runoff water, which occurs due to the significant rain catchment area. In tropical regions, the influencing factor is a sufficiently high level of precipitation, contributing significantly to forming a pit lake. Apart from the presence of runoff water, it is also affected by pumping from nearby pits, although this pumping is not always used in the development of a pit lake. The model we have presented can be an accepted model, with an NSE value of 0.50 and a correlation coefficient of 0.90. This model can apply to pit lakes' predictive development in Indonesia and in other regions, after characterizing the factors of pit lake formation.

**Author Contributions:** Conceptualization, R.S.G., G.J.K. and K.P.; methodology, R.S.G., G.J.K. and E.J.T.; software, E.J.T.; validation, G.J.K., A.A.K. and E.J.T.; formal analysis, R.S.G., G.J.K. and E.J.T.; investigation, E.J.T., K.P. and Y.P.; resources, R.S.G. and K.P.; data curation, E.J.T. and Y.P.; writing—original draft preparation, E.J.T.; writing—review and editing, R.S.G., G.J.K. and A.A.K.; funding acquisition, R.S.G., G.J.K. and K.P. All authors have read and agreed to the published version of the manuscript.

**Funding:** This research was funded by the Bandung Institute of Technology through the Mining Engineering Research Group (KK-TA) at the Faculty of Mining and Petroleum Engineering from a research grant as part of the program titled: Program Penelitian, Pengabdian kepada Masyarakat dan Inovasi (P3MI) Kelompok Keahlian ITB 2020 and supported by The Environmental Department of PT. Kaltim Prima Coal.

**Acknowledgments:** We are very grateful to the management of PT. Kaltim Prima Coal, especially HSE Manager Imanuel Manege and Geology Manager Munir Zein for their support in carrying out this research.

**Conflicts of Interest:** The authors declare no conflict of interest.

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
