# Peer review of "Water Balance of Pit Lake Development in the Equatorial Region"

_water, doi:10.3390/w13213106_

Round 1

Reviewer 1 Report

This study simulated the water balance of a pit lake by a Rainfall-Runoff NRECA Hydrological Model and a balance model for the pit. Overall, I feel this paper lacks originality. The models that the authors used are very simple, and the simulation results are not outstanding. This paper does not fit well with the scope of the special issue that focuses on new methods and algorithms for hydrological modeling.

In addition, the paper is poorly written with grammar errors throughout the paper (e.g. L48, L51, L65, L67, L114, L142, L154, L173, L207, L208, L220, L225, L234, L246, and etc.). The authors misuse active and passive voices many times. Some sentences are really difficult to understand.

In the following are some specific comments:

  1. The introduction section fails to review relevant studies and points out the novelty of this study.
  2. L150 Why do the authors choose to use the empirical NRECA model? Is this model suitable for tropical conditions?
  3. L159-L160 What value?
  4. L170 What value?
  5. L187 The authors assume that potential ET is equal to actual ET. I do not think this assumption will hold, and this seriously compromises the validity of their model.
  6. Figure 4: The authors fail to discuss how they derive surface area at different water level.
  7. Table 1 / Table 2 and Figure 8 overlap.
  8. L293-294 The numbers do not match what is shown in figure.
  9. Figure 13 and Figure 14 seem to overlap.
  10. L399 Figure 15 is not shown.
  11. There is a lack of in-depth discussions of the results and implications of the study.

Reviewer 2 Report

  1. The manuscript is concerned with Hydrological Model of Pit Lake Development in Equatorial Region, which is interesting. It is relevant and within the scope of the journal.
  2. However the manuscript, in its present form, contains several weaknesses. Appropriate revisions to the following points should be undertaken in order to justify recommendation for publication.
  3. Poor grammatical construction throughout the manuscript. Example from line 31 to 39 of the manuscript is full of grammatical mistakes, same as the abstract and several other parts of the manuscript. In line 42 a sentence is actually started with a small letter. In fact from the introduction to the conclusion, the grammatical errors are more than a hundred.
  4. The authors should define the modeling steps more methodically.
  5. Properly explain output metrics including water concentration, turnover rate, outflow rate or water level.
  6. Implement quality assurance procedures. Screen input data for outliers, unit conversion errors, analytical or instrument malfunctions, and other perennial pitfalls. An important consideration in geochemical models is the charge balance of the input solutions; as a general best practice, input solutions should be charge-balanced prior to model simulations with geochemical speciation software.
  7. The authors failed to calibrate and validate results. As the lake is in the filling stage, the authors must compare model predictions with observed data.
  8. The authors did not conduct an uncertainty or sensitivity analysis. They should quantify confidence limits and identify key sensitivities.

Reviewer 3 Report

It is an interesting research topic. It aims to a water balance model for a pit lake formed by open coal mining and continuously affected by pumping and backfilling. The authors combine the NRECA model for calculating the surface and groundwater flow and a water budget equation for the water balance over the lake. This simply methodology can largely solve the problem. The conclusions drawn from this study could have the potential to improve the understanding on how natural and anthropogenic processes influence the hydrological processes of pit lake and benefit the management. However, there are some serious flaws in the current manuscript and it cannot be published until those flaws get resolved after a rewrite. I encourage the authors resubmit this paper after a thorough revision.

  1. Manuscript’s structure is problematic. I suggest the authors reorganize the structure. For example, the study area topography, characteristics, and precipitation/pumped water conditions currently in the Results section should be moved to the Materials and Methods section. Meanwhile, most contents of the current Discussion are more suitable to be results.
  2. As the results show that surface runoff and groundwater supply from the encompassing catchment are the main contributions to water balance for the pit lake, which are simulated by the NRECA model, no information about how the NRECA model is driven, calibrated and validated are provided in the manuscript. In addition, we also concern (1) What is the extent of the catchment? (2) Necessary calibration/validation for the model to make sure the NRECA model works as expected.
  3. PM equation is not good at for estimating lake PET. Please use other empirical formulas more suitable for lake. There are many studies related to lake evaporation estimation.
  4. The description on the proposed model is clearly not adequate. For example, in water balance equation (eq.2), no pumping component is included but it is a unique and very important characteristic of this pit lake. How to address the effects of backfilling in the model is also absent from this part. Overall, we would like to see a model specific to your problem in the model subsection.
  5. When the lake water level becomes higher than groundwater table of soils around the lake, the lake will recharge ground water. The current model lacks consideration of this interactions, likely leading to incorrect estimates of groundwater supply in the later stage of the pit lake development.
  6. Overburden backfilling not only affects the area of the lake and thus evapotranspiration, but also affects the actual storage and groundwater flow by changing the vertical geometry and topography of the lake. Those effects are not considered in the current model. Also, I concern how to estimate the lake storage based on observed water level and extents from satellite in your current methodology.
  7. Existing lake hydrological modeling studies should be included as part of the introduction.
  8. Line 399, I cannot find Figure 15.

Reviewer 4 Report

The article deals with interesting analysis of an open pit lake water balance taking into account a pit lake filling process in simultaneous backfilling conditions. Appropriate tests and investigations were carried out in the mining area of PT. Kaltim Prima Coal located in the East Kalimantan Province, Indonesia, to explore hydrological components like ET, precipitation, water inflows, pumping. A sufficient data for the period of 2014-2017 have been considered and being used in simplified lumped parameter hydrological model. So overall the results seem to be supported enough by data, having in mind that model calculations to achieve general water balance are based on very simplified assumptions. It can be assumed that despite such simplifications the methods are correctly used to obtain convincing results.

Due to so simplified parameter model the title should be just : Water Balance of Pit Lake Development in Equatorial Region.

In the introduction is mentioned the Łuk Mużakowa which occurs in Poland and not as stated in Turkey (!) please read carefully in reference (9).

In description of study site some names are given which should be shown in the figure (Sangatta; Bengalon and Sangatta rivers; Dome area; the Makassar Strait??)

Data about rainfall or pumping water should be presented earlier in data description, not in the results. There is some confusion about the use of terms evapotranspiration and evaporation, especially in the chapter Potential Evapotranspiration or fig. 8, but necessary to check in the entire text.

The conclusions have to be further elaborated and expanded.

There are many different errors and mistakes in the text that need to be checked, including the English grammar/spelling e.g. “This model expects can be used...”, “....Model were used...”; “...which indicate took part to influence...”; “...was influence both capacity of...”; “....water filling was comes despite ....”; “....an elevation position at a specific elevation....”; “The Surface water ....”; “....It can see that...” and others.

My suggestion is to give more information about the study site, especially about geological and hydrogeological conditions surrounding pit area.

The figures are of good quality and legible, however some improvements must be considered. The fig. 7 is not necessary and should be deleted. No descriptions are legible in the figure 4 or fig. 6. Fig. 14 – superscript in m3, what is R-PET? What the red line means in the fig. 1? – it looks like model grid boundary, but numerical model was not presented.

Line 115- de-velopment

Line 135 – subscripts must be the same as in the text

Line 160 – difference greater than 200 m ??

Line 380 – superscript in m3.

The paper is quite clearly written, with sufficient explanation of the methods. The problem was solved and the conclusions are presented. However, the conclusions should contain some comments about water balance of the pit lake. So, more attention should be paid to summing up the results.

All the references seem to be sufficient, however the list should be supplemented with examples of advanced numerical modelling methods applied for similar artificial lake problems like: El-Zehairy A. A., Lubczynski  M. W., Gurwin J., 2018: Interactions of artificial lakes with groundwater applying an integrated MODFLOW solution. Hydrogeol J. 26 (1): 109-132, DOI 10.1007/s10040-017-1641-x

So, please consider the suggestions given above.

Round 2

Reviewer 1 Report

  • I have to repeat my previous comments. The models that the authors used are very simple, and the simulation results are not outstanding. This paper does not fit well with the scope of the special issue that focuses on new methods and algorithms for hydrological modeling.
  • The texts marked red need extensive language editing.
  • L164-L165: Three or four parameters were used in this model to generate rainfall data on the basis of monthly discharge data?
  • The authors seem to calculate ET based on Meyer’s Formula. How did they obtain the values of actual vapor pressure? If they use Meyer’s Formula, what is use of calculating PET then?
  • L207: What software?
  • Water loss is due to actual evaporation not potential evaporation.
  • L481-482: Values between 0.0 and 1.0 are generally viewed as acceptable levels of performance, whereas values < 0.0 indicates unacceptable performance?
  • I do not understand what are the lines in Figure 16. I could not understand the authors’ sensitivity analysis either.
  • There are too many figures in the paper, and much overlapping in different sections of the paper.

Reviewer 2 Report

The author of this article has revised it according to the revision suggestions.The paper is acceptable for publication.

Reviewer 3 Report

I appreciate the revisions made by the authors. Some of my concerns have been resolved/explained but some still do not. I think they are very important. I hope the authors can seriously consider them.  

  1. The manuscript structure still needs to be further improved. In general, Results section should contain all the experimental results produced by this study. However, in current version there are still a large part of important model results in Discussion Section rather than in Results..
  2. It is still not clear on how to estimate the lake water volume according to lake surface area and water level under the condition that the lake geometry is changed by backfilling. Please strength this point in the manuscript. I mentioned this point in previous comments but the authors neglect this point in the response letter.
  3. The lake volume is simultaneously influenced by both surface/groundwater runoff and lake precipitation/evaporation. Thus, it is not a good solution to validate the surface runoff and groundwater discharge from NRECA model using the total lake volume. Please consider to enhance the model validation. I also mentioned this point previously.

According to above comments, I recommend that a major revision is still required before consideration of publication.
